# Comparative effectiveness of glucagon-like peptide-1 receptor agonists on body composition and anthropometric indices: A protocol for a systematic review and network meta-analysis of randomized controlled trials

**Nuttaya Wachiraphansakul[1], Thanawat Vongchaiudomchoke[1], Worapaka Manosroi[2], Surapon Nochaiwong[3,4]** *

1 Department of Internal Medicine, Lampang Hospital, Lampang, Thailand, 2 Division of Endocrinology, Department of Internal Medicine, Faculty of Medicine, Chiang Mai University, Chiang Mai, Thailand, 3 Pharmacoepidemiology and Statistical Research Center (PESRC), Chiang Mai University, Chiang Mai, Thailand, 4 Department of Pharmaceutical Care, Faculty of Pharmacy, Chiang Mai University, Chiang Mai, Thailand

* surapon.nochaiwong@gmail.com

## Abstract

### Background

To date, no studies have addressed the comparative efficacy of glucagon-like peptide-1 receptor agonists (GLP1-RAs) therapy on body composition and anthropometric indices among adult overweight or obese patients with or without type 2 diabetes. To provide evidence-based recommendations, we will conduct a traditional pairwise and network meta-analysis of all available randomized clinical trials that evaluated the effects of GLP1-RAs interventions for adult overweight or obese patients with or without type 2 diabetes.

### Methods and design

Electronic databases, including Medline, Embase, PubMed, Cochrane Library (CENTRAL), Scopus, and CINAHL, will be searched from inception without language restriction. Grey literature will be searched, including Google Scholar, ongoing clinical trial registries, and preprint reports. Reference lists of included trials, relevant major endocrinology scientific meetings, and manual hand searches from key general medicine and obesity and endocrinology journals will also be browsed. Two authors will screen, select, extract, appraise the risk of bias, and rate the evidence findings. Any disagreement will be resolved through team discussion. Based on a random-effects model, we will employ a two-step approach of traditional pairwise meta-analysis and network meta-analysis for quantitative synthesis. The pooled effect estimates using a frequentist approach with 95% confidence intervals for continuous endpoints will be expressed as the standardized mean difference, whereas odds ratios will be used for categorical endpoints. The quality of included trials will be evaluated using the Cochrane risk-of-bias version 2 assessment tool. Certainty of evidence for each

**Data Availability Statement:** No datasets were generated or analysed during the current study. All relevant data from this study will be made available upon study completion.

**Funding:** The authors received no specific funding for this work.

**Competing interests:** The authors have declared that no competing interests exist.

outcome will be assessed using the modified confidence in network meta-analysis approach and the Grading of Recommended Assessment, Development, and Evaluation approach. The magnitude of the effect size, prediction intervals, surface under the cumulative ranking curve values, and certainty of evidence will be incorporated to draw evidence-based conclusions.

## Conclusion

This systematic review and network meta-analysis will summarize the comparative efficacy of GLP1-RAs therapy on body composition and anthropometric indices. Evidence identified from this review will promote the rational use of interventions for adult overweight or obese patients with or without type 2 diabetes and will serve as an important step for evidence-based practice within this area.

## Trial registration

**PROSPERO registration number:** CRD42023458228.

## Introduction

Being overweight or obese is associated with a subsequent risk of adverse health outcomes throughout the life course [1–4]. Globally, the number of adults with obesity has been continuously increasing, from 69 million in 1975 to 390 million in 2016 for women and from 31 million in 1975 to 281 million in 2016 for men [5]. Moreover, 1.30 billion adults were in the range of overweight [5]. Although body mass index (BMI) provides the most useful tools in diagnosing overweight or obese conditions, it does not distinguish body composition, particularly fat distribution, which has been established as a determinant of metabolic risk and related to cardiometabolic-based chronic disease [1, 3]. Indeed, body fat or adipose tissue can be classified based on different anatomical localization. Theoretically, higher visceral adipose tissue or ectopic adipose tissue leads to increasing immune cell infiltration and secretion of vasoconstrictor mediators, resulting in insulin resistance, vascular dysfunction, and cardiovascular disease [6].

In fact, not only controlling overall body weight or promoting weight loss, treatment intervention that supplemented further benefit by decreasing fat distribution is currently a favorable agent for overweight or obese persons as it directly modifies the progression of cardiometabolic-based chronic disease continuums [1, 6]. To date, promising pharmacologic treatment options for overweight or obese persons have become available, particularly a class of medication that improved glucose-mediated insulin secretion—namely glucagon-like peptide-1 receptor agonists (GLP1-RAs) [7]. Besides glycemic control and promotion of weight loss among diabetes and non-diabetes populations [7–12], existing clinical trials and previous systematic reviews also illustrated that GLP1-RAs have further benefits on change in body composition and anthropometric indices (i.e., visceral, subcutaneous, epicardial fat mass, BMI, and waist circumference) [8, 12–15].

However, there are several limitations regarding the currently available systematic review and meta-analysis of GLP1-RAs among adult overweight or obese persons. These include the following: (i) most of existing reviews focused mainly on glycemic control and weight reduction, in which not comprehensive to body composition and anthropometric indices outcome

[12, 16, 17]; (ii) the effect estimates are based on the synthesis of a combination of medication class therapy (pairwise meta-analysis) rather than comparison among individual GLP1-RAs therapy [13–15]; and (iii) studies being based on mixed populations or focused on particular population (i.e., nonalcoholic fatty liver disease or polycystic ovary syndrome) [8, 9, 11, 12, 15]. In this regard, no systematic review has compared individual GLP1-RAs therapy or different dosage regimens among overweight or obese persons with or without type 2 diabetes. To close this knowledge gap and inform rational use of treatment for overweight or obese populations, we planned to perform a systematic review and network meta-analysis of all available randomized clinical trials to summarize and compare evidence regarding individual GLP1-RAs therapy on body composition and anthropometric indices among adult overweight or obese patients with or without diabetes.

## Methods

This systematic review and network meta-analysis will be conducted in compliance with the Cochrane Collaboration Handbook for Systematic Reviews of Interventions [18]. The protocol was registered in the International Prospective Register of Systematic Reviews (PROSPERO) and is currently available online (http://crd.york.ac.uk/prospero, registration number CRD42023458228). The reporting protocol followed the guidelines of the Preferred Reporting Items for Systematic Review and Meta-Analysis Protocols (PRISMA-P) statement (S1 Table) [19].

### Data sources and search strategy

A systematic search will be performed using standard electronic biomedical databases, including Medline, Embase, PubMed, Cochrane Library (CENTRAL), Scopus, and CINAHL. The search for relevant evidence will be conducted from the inception dates of each database with no language restrictions. A search strategy will be constructed in collaboration with an experienced medical librarian. The search terms will be constructed based on a combination of main keywords or Medical Subject Headings terms. The pre-specified searching strategy and the results of the preliminary searches for each database are provided in S2 Table. We will also perform a search through Google Scholar, ongoing clinical trial registries, and preprint reports for grey literature (S3 Table). An additional targeted manual search will be conducted to identify other eligible trials by searching the reference lists of the retrieved clinical trials, relevant major endocrinology scientific meetings, and key general medicine and obesity and endocrinology journals.

### Study selection process and eligible criteria

Records identified from each database will be combined and deduplicated using a citation manager. Two investigators (NW and TV) will screen these records independently using the Rayyan platform, a web application for systematic reviews [20]. Any discrepancy during the screening process will be resolved by consulting the clinical experts (WM) and methodologists (SN). We will include all randomized clinical trials that investigated the clinical efficacy of GLP1-RAs therapy on body composition and anthropometric indices for overweight or obese adults with or without type 2 diabetes, with a follow-up period of at least six months. The pre-specified possible network intervention nodes included in this systematic review and network meta-analysis will be albiglutide, dulaglutide, efpeglenatide, exenatide, liraglutide, lixisenatide, semaglutide, taspoglutide, and tirzepatide. The specific criteria for inclusion and exclusion of relevant studies in terms of PICOTS (population, intervention, comparator, outcome, time frame, and study design) are provided in Table 1.

**Table 1. Study inclusion/exclusion criteria.**

| Study Elements | Criteria for Inclusion | Criteria for Exclusion |
|---|---|---|
| Populations | Adult overweight or obese participants aged ≥ 18 years with or without type 2 diabetes | • Studies that recruiting participants aged less than 18 years or participants with history of type I diabetes, NAFLD, PCOS, or undergoing bariatric surgery<br>• In vitro, in vivo, or animal studies |
| Interventions | • Pharmacologic treatment with GLP1-RAs or dual GIP/GLP1-RAs in different dosages and any route of administration | • Studies with disconnected node of treatment interventions |
| Comparators | • Placebo, active comparator, or standard of care | • Studies without control groups (single arm studies) |
| Outcomes | • Primary outcomes<br> ❖ Change in body fat distribution, visceral adipose tissue, and subcutaneous adipose tissue from baseline measured by validated methods (i.e., DXA, MRI, or CT scan) [45, 46]<br> ❖ Change in skeletal muscle mass or lean body mass from baseline measured by validated methods (i.e., DXA), MRI, CT scan, or BIA) [45, 46]<br>• Secondary outcomes<br> ❖ Change in anthropometric indices from baseline, including body weight, BMI, waist circumference, and waist-to-hip ratio<br> ❖ Unacceptability of treatment: study withdrawal due to any cause<br> ❖ Tolerability of treatment: study discontinuation due to adverse event<br> ❖ Incidence of serious adverse events: participant with at least one reported serious adverse events<br> ❖ Incidence of treatment-emergent adverse events: participant with at least one reported adverse event<br> ❖ Total gastrointestinal adverse events: participant with at least one reported gastrointestinal adverse event<br>• Additional outcomes<br> ❖ Glycated hemoglobin (HbA1C)<br> ❖ Insulin resistance index: HOMA-IR<br> ❖ Treatment adherence<br> ❖ Psychosocial aspects (e.g., depressive symptoms, anxiety, distress/stress, and well-being)<br> ❖ Patient-reported health-related quality of life<br> ❖ Other patient-reported outcomes (e.g., treatment satisfaction)<br> ❖ Health care costs | • Studies with unclear/invalid methods of measurement of body composition and anthropometric indices<br>• Studies not providing data to calculate the effect estimates of the outcome of interest<br>• Studies with a follow-up period of less than 6 months |
| Time frame | • From the inception dates of each databases to current (an updated search will be conducted before formal analyses) | • No restrictions were imposed on timing of start date or language |
| Study design | • Published full article: parallel randomized clinical trials | • Crossover trials, quasi-experimental studies, N-of-one, case series/case reports, observational studies (cohort, case-control, cross-sectional), pharmacokinetic/pharmacodynamics studies, and qualitative studies<br>• Reports not involving original data (i.e., narrative review, systematic review, meta-analysis, news items, consensus statement, guidelines, and opinion/editorials) |

Abbreviations: BIA, bioelectrical impedance analysis; BMI, body mass index; CT, computerized tomography; DXA, dual X-ray absorptiometry; GIP, glucose-dependent insulinotropic polypeptide; GLP1-RAs, glucagon-like peptide-1 receptor agonists; HOMA-IR, Homeostatic Model Assessment for Insulin Resistance; MRI, magnetic resonance imaging; NAFLD, nonalcoholic fatty liver disease; PCOS, polycystic ovary syndrome.

## Data extraction

Two investigators (NW and TV) will perform an independent data extraction using a standardized approach and electronic extraction form. The following data will be collected from each clinical trial based on:

i. Trial characteristics (i.e., the name of the first and the corresponding authors, study year, study location, study setting [single-center, multicenter], types of design [placebo-controlled or active-controlled], study population [inclusion and exclusion criteria], study size of each treatment group, and follow-up period)

ii. Participant baseline characteristics (i.e., age, sex, race/ethnicity of the participants, body weight, body mass index, waist circumference, blood pressure, previous treatment, comorbid conditions [e.g., type 2 diabetes, hypertension, kidney disease, obstructive sleep apnea, symptomatic osteoarthritis of the knee], baseline laboratory parameters (i.e., hemoglobin A1c, fasting plasma glucose, lipid profiles, c-reactive protein, estimated glomerular filtration rate, plasminogen activator inhibitor-1])

iii. Treatment intervention (i.e., specific GLP1-RAs or glucose-dependent insulinotropic polypeptide [GIP]/GLP1-RAs treatment, comparator treatment, route of administration, dosage, frequency of treatment, and concomitant medication)

iv. Predefined outcomes of interest (primary, secondary, and additional outcomes), including their measurement methods in detail

Data extraction will be cross-checked by two investigators (WM and SN). Any discrepancies during the extraction process will be resolved through a group discussion. For continuous endpoints (i.e., body composition and anthropometric indices from baseline), the mean and standard deviation (SD) will be extracted. If numerical data are not provided, we will extract the relevant data from available figures using the WebPlotDigitizer 4.4. In the case where the mean and SD are not provided, we will estimate the sample mean with SD using the sample size, median, interquartile range, and/or range (min-max) [21]. Ultimately, we will employ to estimate treatment effect based on mean change from baseline for continuous outcomes to address the different measurement methods and baseline participant characteristics across included trials. For any categorical endpoint, 0.5 cells will be applied for treatment arms with zero events [22]. For trials with missing or unclear data, we will contact the corresponding author via e-mail for further clarification. If the corresponding author does not reply within two weeks, a second attempt will be made. If there is no response after the second attempt, the data will be excluded from the analyses or using the most relevant reported data, as appropriate.

## Quality assessment

All included studies will be independently evaluated in terms of their quality using the Cochrane risk-of-bias version 2 assessment tool (RoB 2) [23] by two investigators (NW and TV). The RoB 2 evaluates the presence of potential biases in randomized clinical trials based on five domains, including the randomization process, deviations from the intended interventions, missing outcome data, measurement of the outcome, and selection of the reported result [23]. Each study will be rated as low risk, some concerns, or high risk. During the rating process, any discrepancies will be reached by discussion with the clinical experts (WM) and methodologists (SN).

## Data synthesis

The justification for conducting the formal data synthesis will be evaluated using tabulation methods, where the characteristics of all included studies are listed and compared. However, if there is limited data on the included trials, we will perform a systematic review and narrative synthesis regarding the key participant characteristics and treatment comparisons.

For each pairwise comparison, all studies will be assessed in terms of both clinical and methodological heterogeneity. Statistical heterogeneity will be evaluated using the Cochran $Q$ test with a $P$-value of less than 0.10. The degree of inconsistency will be evaluated using $I^2$ and $\tau^2$ statistics. Transitivity will also be assessed for network comparison, which looks for the similarity between treatment comparisons. Studies that are too different from the rest will be excluded.

A two-step approach for quantitative syntheses will comprise conventional pairwise meta-analysis and network meta-analysis. For the first step, conventional pairwise meta-analysis will be performed for each pairwise comparison using a random-effects model regardless of the degree of statistical heterogeneity [24]. Next, a network meta-analysis will be conducted to estimate the comparative efficacy for each outcome of interest among available GLP1-RAs treatment interventions using a frequentist approach with restricted maximum likelihood estimation. A network geometric plot will be visualized to evaluate the pattern of the connected nodes for each outcome of interest. Network estimates multivariate modeling will be executed using a consistency model. The inconsistency assumption will be tested using the global test or Cochran's $Q$ statistics, loop inconsistency, and node-splitting approach [25]. The results of both the consistency and inconsistency models will also be compared. The surface under the cumulative ranking curve (SUCRA) will be estimated and used to rank the treatment interventions within the connected network [26]. Rankograms will also be used to visualize the predicted probability of being the best for each treatment [26]. Where the included trials are at least ten trials for each outcome, the presence of potential small study effects will be assessed using comparison-adjusting funnel plot symmetry [26].

A set of subgroup analyses will be planned to examine the changes in the comparative treatment effects across different levels of the following effect modifiers:

i. Trial characteristics (i.e., sample size [<100, 100–500, or >500 participants], trial setting [single-center vs. multicenter], duration of treatment follow-up [6–12 vs. >12 months], study quality based on the RoB 2 assessment [low, some concerns, or high], and geographical regions)

ii. Participant characteristics (i.e., age [young adults 18–34, middle-aged 35–65, and elderly aged ≥65 years], sex [male vs. female], race/ethnicity, history of type 2 diabetes [yes vs. no], other comorbid conditions, and baseline BMI and hemoglobin A1c [<7.0% vs. ≥7.0%)

Additionally, network meta-regression analyses will also be planned by supplementing other covariates (both trial and participant characteristics), as mentioned in the adjusted network meta-analysis model. A series of pre-specified will also be applied to assess the robustness of the findings under the following conditions: (i) removing a single study one at a time (a leave-one-out approach); (ii) removing studies with a high risk of bias; (iii) removing studies with a small study size (i.e., less than 25th percentile); (iv) removing studies published before 2016; (v) performing separate analyses for active-controlled trials and placebo-controlled trials; and (vi) adding data from unpublished literature (i.e., conference abstracts, thesis, proceedings).

We will summarize the effect estimates as standardized mean differences for continuous endpoints and odds ratios for categorical endpoints, with corresponding 95% confidence intervals. Furthermore, the 95% prediction intervals for all pooled estimates will be calculated and presented concordantly [27]. The network estimate for specific treatment comparisons will be presented using forest plots and league tables. In addition, a rank-heat plot will be visualized to sum up the SUCRA values of multiple outcomes of interest. All analyses will be performed using Stata 16.0 (StataCorp, College Station, TX, USA). Analysis results with a two-tailed $P$-value of less than 0.05 will be considered statistically significant.

## Certainty assessment and classification of interventions

Two investigators (NW and SN) will perform an independent grading of certainty and rating of the evidence for each outcome of interest using the modified confidence in network meta-analysis approach [28] and the Grading of Recommended Assessment, Development, and

Evaluation approach [29]. Overall, the upgrading or downgrading the quality of evidence depends on the risk of bias, imprecision, inconsistency, and indirectness of the findings. For judging evidence certainty, we will categorize the strength of body evidence as very low-, low-, moderate-, and high-quality [30, 31]. Any disagreement regarding evidence certainty assessment will be resolved through a team discussion.

To summarize the network effect estimates and classification of interventions, a contextualized approach based on clinical and methodological aspects will be made for each treatment comparison. Specifically, this approach will incorporate the magnitude of the treatment effect size, prediction intervals, certainty of evidence, and SUCRA values to draw an evidence-based conclusion for each treatment comparison [31–33]. Finally, each treatment comparison will be classified as having trivial (neutral effect compared to placebo/standard of care/usual care), small, moderate, or large effects on body composition and anthropometric indices for overweight or obese persons with or without type 2 diabetes.

## Ethics and dissemination

Ethical approval is not required because this work will use data based on published literature and does not directly involve human subjects. Findings from this systematic review and network meta-analysis will be reported in compliance with the Preferred Reporting Items for Systematic reviews and Meta-Analyses (PRISMA) 2020 statement guidelines [34] and the PRISMA extension statement for reporting systematic reviews incorporating network meta-analysis of healthcare interventions [35]. Our findings will be published in peer-reviewed journals. The final report will address any further amendments to the review protocol.

## Discussion

Pharmacologic interventions for overweight or obesity produce greater weight loss than lifestyle modification alone and should be considered in all overweight or obese persons with weight-related complications [2]. GLP1-RAs are a class of medication that targets incretin hormone action, and its receptors are widely distributed in the brain, heart, islets, and other organs. Several clinical trials have demonstrated that GLP1-RAs effectively treat obese persons [36]. Regarding landmark clinical trials, a study of daily subcutaneous injection of liraglutide 3.0 mg produced mean weight losses of 7.8 kg, compared with 2.0 kg in the placebo-treated group [37]. A 2.4 mg weekly injection of semaglutide also resulted in a mean weight loss of 15 kg from baseline compared to 2.6 kg in the placebo group [38]. Given the efficacy of weight loss shown from these clinical trials, 3 mg daily of liraglutide and 2.4 mg weekly semaglutide are now approved by the United States Food and Drug Administration (US FDA) for anti-obesity medication. Liraglutide is also mentioned in part of pharmacotherapy of obesity management from both the United States and European guidelines for treating obesity in long-term use [2, 3, 39]. Tirzepatide—a dual action, once-weekly GIP/GLP1-RA, also revealed significant weight loss in obese patients with or without type 2 diabetes [40, 41]. A 15 mg weekly of tirzepatide subcutaneous injection resulted in a reduction of 23.6 kg after 72 weeks of study treatment [40]. To date, tirzepatide is now under investigation process for anti-obesity medication from the US FDA, with an expectation to be approved by the end of 2023.

Multiple mechanisms of GLP1-RAs for bodyweight reduction have been identified, including decreased appetite and hunger, increased satiety, delayed stomach emptying time, and decreased calorie intake [42]. Although the most widely used anthropometric indices in clinical practice and clinical trial endpoints indicate the efficacy of weight-loss medication are body weight and BMI, monitoring of excessive degree of adiposity after treatment is limited by the availability and costs of imaging techniques for body composition assessment [43]. Assessment

of adiposity is also one of the useful markers of cardiometabolic disease. Evidence supports that a decrease in visceral fat tissue can ameliorate obesity-related metabolic complications [1, 6]. Previous systematic reviews found that GLP1-RAs significantly reduced adiposity (i.e., visceral, subcutaneous, and epicardial fat) [13–15]. However, these systematic reviews did not simultaneously compare individual GLP1-RAs therapy or different dosage regimens among overweight or obese persons with or without type 2 diabetes [13–15].

To address this knowledge gap, we will further compare the magnitude of adiposity reduction between individual GLP1-RA and/or different dosages. Although there is no definite cut-off level for change in the percentage of body composition to predict health risk, our findings will lead to future cardiovascular outcome studies of potential doses and/or particular GLP1-RAs focusing on the change of body composition. Theoretically, GLP1-RAs that have cardiovascular benefits and provide additional effects on visceral adiposity may be considered and have promising treatment in clinical settings, particularly for those with existing or higher risk of cardiovascular disease. Furthermore, another issue that must be considered is muscle mass. Indeed, a relative loss of lean muscle mass, the so-called "sarcopenic obesity", characterized by excessive adiposity and reduced lean muscle mass or muscle strength, significantly contributes to health risks [44]. On the contrary, weight reduction methods may lead to enhanced muscle loss. Meanwhile, evidence regarding the effects of GLP1-RAs on muscle mass is currently limited. As such, we postulate that our findings will yield additional, comprehensive evidence regarding the use of GLP1-RAs on body composition and anthropometric indices, which could be a part of improving the patient's care management and health-related quality of life.

In this circumstance, managing overweight or obese persons with or without type 2 diabetes remains challenging for long-term outcomes and sustainability of treatment over time, which raises concerns about the optimality of the recommendations. A systematic and comprehensive review of current clinical evidence regarding GLP1-RAs therapy is necessary to provide overall comparative effectiveness of individual treatment options for adult overweight or obese patients with or without type 2 diabetes. To our knowledge, this systematic review and network meta-analysis will be the first to summarize individual GLP1-RAs therapy on body composition and anthropometric indices in this group of patients. Our review will be conducted using a rigorous approach without language restrictions to include all available evidence. We anticipate that our findings will influence international guideline recommendations and improve the treatment of care and well-being of overweight or obese persons.

## Conclusions

This systematic review and network meta-analysis will summarize the comparative efficacy of available GLP1-RAs therapy on body composition and anthropometric indices. Evidence identified from this review will promote the rational use of optimal treatment and inform evidence-based practice for adult overweight or obese patients with or without type 2 diabetes. We anticipate that the results of our review would be of substantial benefit to stakeholders who are involved in this group of patients. The findings from this review will also clarify the existing gaps in evidence and guide researchers to undertake more clinical trials to advance our understanding of the most optimal treatment for overweight or obese persons. We plan to disseminate our complete systematic review through peer-reviewed medical journals.

## Supporting information

**S1 Table. PRISMA-P 2015 checklist: Recommended items to address in a systematic review protocol.**
(PDF)

**S2 Table. Systematic review search strategy.**
(PDF)

**S3 Table. Grey literature from ongoing clinical trial register.**
(PDF)

## Author Contributions

**Conceptualization:** Nuttaya Wachiraphansakul, Thanawat Vongchaiudomchoke, Worapaka Manosroi, Surapon Nochaiwong.

**Methodology:** Nuttaya Wachiraphansakul, Thanawat Vongchaiudomchoke, Surapon Nochaiwong.

**Project administration:** Nuttaya Wachiraphansakul, Thanawat Vongchaiudomchoke, Surapon Nochaiwong.

**Resources:** Nuttaya Wachiraphansakul, Thanawat Vongchaiudomchoke, Surapon Nochaiwong.

**Software:** Nuttaya Wachiraphansakul, Thanawat Vongchaiudomchoke, Surapon Nochaiwong.

**Supervision:** Surapon Nochaiwong.

**Writing – original draft:** Nuttaya Wachiraphansakul, Thanawat Vongchaiudomchoke, Surapon Nochaiwong.

**Writing – review & editing:** Worapaka Manosroi, Surapon Nochaiwong.

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
