## [Decision Letter · Decision Letter 0]

7 Dec 2023

PONE-D-23-30493Comparative Effectiveness of Glucagon-Like Peptide-1 Receptor Agonists on Body Composition and Anthropometric Indices: A Protocol for a Systematic Review and Network Meta-Analysis of Randomized Controlled TrialsPLOS ONE

Dear Dr. Nochaiwong,

Thank you for submitting your manuscript to PLOS ONE. After careful consideration, we feel that it has merit but does not fully meet PLOS ONE’s publication criteria as it currently stands. Therefore, we invite you to submit a revised version of the manuscript that addresses the points raised during the review process.

We look forward to receiving your revised manuscript.

Kind regards,

Saifur Rahman Chowdhury, MPH

Academic Editor

PLOS ONE

Journal Requirements:

**Additional Editor Comments:**

Please consider the followings in the revised version:

1. Follow-up time in table 1 and main text (where applicable)

2. In abstract, mention the analysis method (eg. frequentist approach), mention the RoB assessment tool specifically (eg. RoB 2) and certainty assessment method (GRADE approach)

Reviewers' comments:

Reviewer's Responses to Questions

**Comments to the Author**

1. Does the manuscript provide a valid rationale for the proposed study, with clearly identified and justified research questions?

Reviewer #1: Yes

Reviewer #2: Yes

2. Is the protocol technically sound and planned in a manner that will lead to a meaningful outcome and allow testing the stated hypotheses?

Reviewer #1: Yes

Reviewer #2: Yes

3. Is the methodology feasible and described in sufficient detail to allow the work to be replicable?

Reviewer #1: Yes

Reviewer #2: Yes

4. Have the authors described where all data underlying the findings will be made available when the study is complete?

Reviewer #1: Yes

Reviewer #2: Yes

5. Is the manuscript presented in an intelligible fashion and written in standard English?

Reviewer #1: Yes

Reviewer #2: Yes

6. Review Comments to the Author

You may also provide optional suggestions and comments to authors that they might find helpful in planning their study.

Reviewer #1: The authors present a network meta-analysis protocol, very well constructed and readable, to summarize and compare evidence regarding individual GLP1-RAs therapy on body composition and anthropometric indices among adult overweight or obese patients with or without diabetes.

It was great to review a quality protocol with great potential to provide high-quality evidence when the meta-analysis is completed.

However, I make small suggestions that I believe will leave the manuscript in better condition for publication. In the methodology, authors need to include text on how the risk of bias assessment will be carried out, with a funnel plot if more than 10 studies are included, or with an Egger test if less than ten clinical trials are included. Finally, I suggest that in table 1 under Primary outcomes the authors include the references in which the DXA, MRI, or CT scan methods were validated.

Reviewer #2: In this manuscript, the authors describe their study protocol for a systematic review and meta-analysis of the literature seeking to identify the outcomes of GLP-1RA administration on body composition and anthropometric indices in overweight or obese patients. The authors have provided sufficient introduction, detailed search information, and a strong plan for investigation.

Comments

1. My only comment is regarding the discussion. The authors often mention what new information this review looks to provide, but do not go as far as to discuss the clinical implications of the potential findings. I would encourage the authors to discuss this point further. How would differences in body composition or anthropometric indices change the way we use these medications? Are treatments or indications expected to change? If similar groups of patients would receive these treatments, then the key clinical implications should be described. This will help the reader understand the importance of this review once it is complete, compared to previous reviews on this topic.

7. PLOS authors have the option to publish the peer review history of their article (what does this mean?). If published, this will include your full peer review and any attached files.

Reviewer #1: **Yes: **Ricardo Ney Cobucci

Reviewer #2: No

---

## [Author Response · Author response to Decision Letter 0]

19 Dec 2023

Response to Reviewers

PONE-D-23-30493 entitled, “Comparative Effectiveness of Glucagon-Like Peptide-1 Receptor Agonists on Body Composition and Anthropometric Indices: A Protocol for a Systematic Review and Network Meta-Analysis of Randomized Controlled Trials”

Editor Comments:

Please consider the followings in the revised version:

#1. Follow-up time in table 1 and main text (where applicable)

Thank you very much for your suggestion. We have made changes as recommended in Table 1 and throughout the manuscript. 

#2. In abstract, mention the analysis method (eg. frequentist approach), mention the RoB assessment tool specifically (eg. RoB 2) and certainty assessment method (GRADE approach)

More details regarding the methodological approach for systematic review and network meta-analysis have been addressed in the “Abstract” section as recommended. 

Reviewer 1

The authors present a network meta-analysis protocol, very well constructed and readable, to summarize and compare evidence regarding individual GLP1-RAs therapy on body composition and anthropometric indices among adult overweight or obese patients with or without diabetes.

It was great to review a quality protocol with great potential to provide high-quality evidence when the meta-analysis is completed.

However, I make small suggestions that I believe will leave the manuscript in better condition for publication. In the methodology, authors need to include text on how the risk of bias assessment will be carried out, with a funnel plot if more than 10 studies are included, or with an Egger test if less than ten clinical trials are included. Finally, I suggest that in table 1 under Primary outcomes the authors include the references in which the DXA, MRI, or CT scan methods were validated.

Thank you very much for your insightful comments. Based on the network meta-analysis approach, we have addressed the issues regarding the small study effect in the “Data synthesis” section, as follows:

“Where the included trials are at least ten trials for each outcome, the presence of potential small study effects will be assessed using comparison-adjusting funnel plot symmetry [26].”

As recommended, moreover, references regarding the methods for measurement of body composition were stated. 

References

- Cruz-Jentoft AJ, et al; Writing Group for the European Working Group on Sarcopenia in Older People 2 (EWGSOP2), and the Extended Group for EWGSOP2. Sarcopenia: revised European consensus on definition and diagnosis. Age Ageing. 2019;48(1):16-31. doi: 10.1093/ageing/afy169.

- Lee K, et al. Recent Issues on Body Composition Imaging for Sarcopenia Evaluation. Korean J Radiol. 2019;20(2):205-217. doi: 10.3348/kjr.2018.0479.

Reviewer 2

In this manuscript, the authors describe their study protocol for a systematic review and meta-analysis of the literature seeking to identify the outcomes of GLP-1RA administration on body composition and anthropometric indices in overweight or obese patients. The authors have provided sufficient introduction, detailed search information, and a strong plan for investigation.

Comments

1. My only comment is regarding the discussion. The authors often mention what new information this review looks to provide, but do not go as far as to discuss the clinical implications of the potential findings. I would encourage the authors to discuss this point further. How would differences in body composition or anthropometric indices change the way we use these medications? Are treatments or indications expected to change? If similar groups of patients would receive these treatments, then the key clinical implications should be described. This will help the reader understand the importance of this review once it is complete, compared to previous reviews on this topic.

Thank you very much for your pertinent observation. From a clinical point of view, we totally agree with your observation. Unfortunately, there are no currently comprehensive systematic reviews and meta-analyses to compare the efficacy of glucagon-like peptide-1 receptor agonists (GLP1-RAs) therapy on body composition and anthropometric indices among adult overweight or obese patients with or without type 2 diabetes. Apart from the current evidence of GLP1-RAs on the management of adult overweight or obese patients with or without type 2 diabetes, we postulated that this systematic review and network meta-analysis will provide further benefits of individuals GLP1-RA on body composition. As such, we have revised and addressed more details regarding this issue in the “Discussion” section.

---

## [Editor Report · Decision Letter 1]

8 Jan 2024

Comparative Effectiveness of Glucagon-Like Peptide-1 Receptor Agonists on Body Composition and Anthropometric Indices: A Protocol for a Systematic Review and Network Meta-Analysis of Randomized Controlled Trials

PONE-D-23-30493R1

Dear Dr. Nochaiwong,

We’re pleased to inform you that your manuscript has been judged scientifically suitable for publication and will be formally accepted for publication once it meets all outstanding technical requirements.

Kind regards,

Saifur Rahman Chowdhury, MPH

Academic Editor

PLOS ONE
---

## [Editor Report · Acceptance letter]

17 Feb 2024

PONE-D-23-30493R1 

PLOS ONE

Dear Dr. Nochaiwong, 

I'm pleased to inform you that your manuscript has been deemed suitable for publication in PLOS ONE. Congratulations! Your manuscript is now being handed over to our production team.

Kind regards, 

on behalf of

Dr. Saifur R. Chowdhury 

Academic Editor

PLOS ONE